# Advancing Urban Wastewater Management: Optimizing Sewer Performance through Innovative Material Selection for the Armlet with a Wet Circuit Measurement System

Tomasz Sionkowski [1], Wiktor Halecki [2,*] and Krzysztof Chmielowski [3]

1    Grundfos Pompy Ltd., Klonowa Street 23, 62-081 Przeźmierowo, Poland; tsionkowski@grundfos.com
2    Institute of Nature Conservation Polish Academy of Sciences, Mickiewicza 33, 31-120 Krakow, Poland
3    Department of Natural Gas Engineering, Faculty of Drilling, Oil and Gas, AGH University of Science and Technology, Al. A. Mickiewicza 30, 30-059 Krakow, Poland; krzysztof.chmielowski@agh.edu.pl
*    Correspondence: wiktor.halecki@urk.edu.pl

**Abstract:** Rainwater infiltration presents substantial challenges for urban wastewater management systems. This article delves into enhancing the quality of wastewater systems by proposing a novel device designed to tackle this issue comprehensively. The focal point of our research revolves around the conceptualization, construction, rigorous testing, and the potential multifaceted applications of this innovative wastewater device. Our study is dedicated to assessing the viability of a cutting-edge apparatus that empowers municipal entities in swiftly identifying rainwater ingress points within channels during precipitation events. Our findings vividly showcase the device's remarkable capability to directly measure moisture levels along the channel's path, eliminating the arduous need for manual data input, extensive data collection, and intricate water analysis procedures. To ensure the seamless flow of both sewage and water within the sewer channel, the use of a relatively slender strap is conventionally favored. However, factoring in the requisite structural robustness, we recommend a minimum thickness of 4 mm for 3D printing applications. For instances where maintaining the channel's cross-sectional area integrity is paramount, opting for an armlet with a wet circuit measurement thickness of up to 7 mm is vital. In the realm of material selection, our investigation advocates for the utilization of PC/ABS (polycarbonate/Acrylonitrile Butadiene Styrene), ABS, ASA (Acrylonitrile Styrene Acrylate), or HIPS (High Impact Polystyrene) for strap housing. For sewer diameters surpassing 315 mm, the application of thin-walled PVC (Poly Vinyl Chloride) emerges as a practical recommendation. Notably, the incorporation of PVC flat bars is discouraged, as their presence might potentially hinder the fluidity of sewage flow, thereby compromising the precision of wet circuit measurements. The pivotal innovation lies in the armlet with a wet circuit measurement system, harboring immense potential for broad-scale integration across municipal facilities. This solution emerges as a streamlined and efficient strategy, offering a comprehensive avenue for continuously monitoring, fine-tuning, and optimizing the structural soundness and operational efficacy of sewer systems.

**Keywords:** rainwater ingress; sewer channels; wastewater management; sensor technology

## 1. Introduction

The effective management of urban wastewater systems is a critical aspect of maintaining public health and preserving the environment [1–4]. However, one persistent challenge faced by wastewater management systems is rainwater ingress into sewer channels. Excessive rainwater infiltrating sewer channels can overload the system, resulting in sanitary sewer overflows and environmental pollution [3,5]. Moreover, it places a strain on infrastructure, leading to potential damage and costly repairs. Therefore, the accurate detection and quantification of rainwater ingress are crucial for efficient wastewater management and infrastructure maintenance [6]. Monitoring and managing the wet circuit in closed channels,

such as sewer and stormwater channels, pose significant challenges in today's world of wastewater management [7,8]. There is a need for innovative solutions that enable accurate measurement of the wet circuit in such channels to effectively monitor water flow and identify issues related to rainwater ingress [9]. In addition to rainwater ingress detection, the device also enables proactive maintenance strategies to mitigate the adverse impacts of rainwater infiltration [10]. While this research marks a significant advancement in rainwater ingress detection, there are several challenges that need to be addressed [11–13]. Ensuring the long-term durability and reliability of the device, seamless integration with existing management systems, and scalability for large-scale implementation are areas that require further investigation [14–17]. Future research endeavors may also explore the incorporation of predictive analytics and machine learning algorithms to enhance the device's capabilities for accurate forecasting and intelligent decision-making [18,19]. Timely detection of rainwater ingress allows wastewater management authorities to implement immediate corrective actions, such as targeted cleaning, optimized pump operations, and diversion of excess flows to alternate treatment facilities [20]. These proactive measures contribute to improving system efficiency, reducing treatment costs, and minimizing the risk of overflows and environmental contamination [21–23].

The presence of rainwater in sewer systems can overwhelm the capacity of treatment plants, leading to operational inefficiencies, increased energy consumption, and potential environmental risks [24–27]. Traditional methods of rainwater ingress detection often rely on manual inspections, which are time-consuming and labor-intensive [28]. To address these challenges, the development of a novel device capable of real-time monitoring and proactive maintenance is crucial [29–33]. This paper presents the research efforts towards the advancement of rainwater ingress detection in sewer channels through the development of a cutting-edge device. The primary goal of this research was to develop a novel device capable of detecting and quantifying rainwater ingress in sewer channels. The device aimed to provide real-time monitoring and data analysis, facilitating proactive maintenance and optimizing resource allocation in urban wastewater management systems. By accurately identifying rainwater ingress, the device aimed to enhance system performance and mitigate the adverse impacts of rainwater infiltration. Furthermore, this study sought to investigate different casing materials and shapes for measuring the wet circuit in closed channels, with a specific focus on sewage and rainwater channels. The research findings aimed to inform the selection of suitable casing materials and appropriate shapes to ensure precise and reliable measurements of the moisture circuit. The primary objective was to facilitate efficient control of rainwater and sewage systems, thereby enhancing the overall sustainability and robustness of urban infrastructure. This involved optimizing the components of the moisture circuit, which plays a pivotal role in the effective administration of rainwater and sewage.

## 2. Materials and Methods

### 2.1. Study Object and Data for System Calibration

The material and methods section of this report focuses on the design and construction of the experimental setup for investigating the moisture circuit in closed channels. The experimental setup was carefully designed to replicate the conditions of closed channels, specifically sewage channels and rainwater. Considerations were made to ensure the setup accurately represented the real-world scenarios encountered in these channels. Various instruments and sensors were integrated into the experimental setup to measure and monitor the moisture circuit. This included sensors to measure humidity, water flow, and moisture levels in the channels. The instrumentation was calibrated and validated to ensure accurate and reliable measurements throughout the experiments. The research setup primarily consisted of the following components:

- Supporting structure;
- Lower chamber for sewage or rainwater;
- Pump set for transporting sewage or rainwater;

- Plastic straps;
- Sewage pipes of various diameters;
- Mounting brackets for sewage pipes;
- Power supply system.

*2.2. Investigation of Material Selection for the Plastic Enclosure of the Measurement Strap*

Materials used for constructing the enclosures of the armlet with wet circuit measurements and conducting the tests were selected from various types of plastics, including ABS, Z-ASA, PC/ABS, Z-ULTRAT, Z-HIPS, and PVC. Below, we present the characteristics provided by the manufacturers for each material:

ABS Material

As part of the study, the first set of armlets with wet circuit measurement was made from ABS material. Initially, the enclosures for the straps were designed and then 3D printed.

Z-ASA Pro (Acrylonitrile Styrene Acryl) is a stable thermoplastic material for 3D printing, resistant to changing weather conditions and UV radiation. Z-ASA Pro ensures that models will not change color or physical properties even after extensive testing in open spaces. Its high-temperature resistance allows for testing models under direct sunlight without the risk of deformation or discoloration. Z-ASA Pro is also suitable for precise 3D printing of large models due to its low shrinkage and deformation levels. It is obtained through the copolymerization of SAN (Styrene Acrylonitrile) with acrylic rubber.

PC/ABS is a blend of polycarbonate (PC) and ABS (Acrylonitrile Butadiene Styrene) fibers. The filament is impact-resistant, UV-resistant, and can withstand high temperatures. Z-PCABS is also resistant to various chemicals, including salts, acids, and bases. The material is ideal for prototyping, printing specialized tools, and manufacturing final parts. Objects 3D printed with Z-PCABS remain fully functional and durable even with prolonged use. The filament is also suitable for creating enclosures, tools, structural components, and automotive parts exposed to impact and various chemical substances. Depending on the composition, it combines the properties of both materials. It exhibits good hardness, impact resistance (comparable to PC), resistance to creep under static loads, higher thermal resistance than ABS, good electrical insulation, and low moisture absorption. Below are the selected parameters of the PC/ABS material.

Z-ULTRAT is a 3D printing filament based on ABS (Acrylonitrile Butadiene Styrene) designed to provide excellent strength and model quality (Figure S1). It is ideal for applications where high quality is a priority, such as functional prototyping, final parts, or consumer products. Z-ULTRAT is resistant to high temperatures and impact, and its exceptional hardness allows for demanding tests. It is available in seven colors. The filament is easily processed chemically and mechanically. This material enables the printing of models with physical properties similar to those produced using injection molds, thereby reducing the costs of low-volume production while maintaining high quality. It is obtained through the polymerization of butadiene and the copolymerization of acrylonitrile with styrene, with simultaneous grafting of the resulting copolymer onto polybutadiene. It is an amorphous material with high impact resistance, hardness, and scratch resistance. However, it lacks resistance to light and UV radiation. It exhibits good insulation properties and satisfactory resistance to alkaline substances, diluted acids, aliphatic hydrocarbons, oils, and fats. However, it is not resistant to acids, esters, and ketones. ABS has a wide processing window, especially regarding temperature profiles. Drying of the material is required for applications with high aesthetic surface requirements.

Below are the selected parameters of the ULTRAT material with the following composition: ABS (Acrylonitrile Butadiene Styrene copolymer)—90~100%, PC (polycarbonate)—0–3%.

Z-HIPS (High Impact Polystyrene) is perfect for printing large models. Its strength allows for mechanical and functional testing. Compared to other polystyrene materials, Z-HIPS has been modified to give models a unique semi-matte finish. The surface quality

of prints from this material is comparable to that of components produced using injection molds. With Z-HIPS, it is possible to quickly create prototypes of enclosures and industrial parts without additional post-processing. The material exhibits low shrinkage. It is an amorphous material with good stiffness and surface quality. Depending on the content of butadiene, it can have low or high impact resistance. High-impact types have lower tensile strength and exhibit significantly higher surface matte appearances. Z-HIPS has low moisture absorption, tendency to crack under stress, and susceptibility to electrostatic charging. It has good insulation properties and satisfactory resistance to water, alkaline compounds, diluted inorganic acids, and aqueous solutions of most salts. However, it is sensitive to long-term exposure to atmospheric factors.

*2.3. Problem of Shrinkage in 3D Printing of Armlet with Wet Circuit Measurement*

Three-dimensional printing using Fused Deposition Modeling (FDM) technology, also known as Fused Filament Fabrication (FFF) or Layer Plastic Deposition (LPD) in the case of the ZORTRAX brand (ZORTRAX, Olsztyn, Poland), poses certain challenges. FDM technology was developed by the American-Israeli company Stratasys (Prairie, MN, USA; Rehovot, Israel) in 1988, and the patent for this technology expired in 2009, allowing other companies to use and further develop it. FDM is the most commonly used 3D printing technology, which involves extruding heated thermoplastic material through a print head.

One of the issues that can occur during 3D printing, depending on the material used, is material shrinkage. This shrinkage negatively affects the entire process, and typically, the parts where this phenomenon occurs are not suitable for use. The main reason for shrinkage is the cooling of the printed layers, which causes detachment from the print bed and prevents the correct printing of subsequent layers. This results in warped models and deviations from the intended dimensions, particularly in the Z-axis. When this problem arises, it is usually necessary to abort the print. This problem is most commonly encountered with large models printed with ABS. However, using different materials or ABS with additives that minimize or eliminate shrinkage can help mitigate the issue.

In addition, PVC straps in the form of flat bars were tested during the research. PVC is an interesting and relatively inexpensive material that can be used for constructing enclosures for potential armlets with wet circuit measurement in closed channels. The strap enclosure consists of a PVC flat bar with corner brackets attached to its ends. The purpose of these brackets is to provide the necessary pressure for the armlet with a wet circuit measurement system against the walls of the sewer pipe. Additionally, silicone cones can be used, which are inserted between the corner brackets to increase the stability of the strap attachment in the sewer pipe. Another interesting material for building measurement straps was found to be thin-walled PVC. This solution is recommended, especially for larger sewer pipe diameters where using 3D-printed straps would be too costly. To ensure the seamless flow of both sewage and water within the sewer channel, the use of a relatively slender strap is conventionally favored. Considering the essential structural strength, we advise a minimum thickness of 4 mm for 3D printing applications. In situations where preserving the cross-sectional area integrity of the channel is of utmost importance, opting for an armlet with a wet circuit measurement thickness of up to 7 mm is vital.

*2.4. Experimental Procedures*

To validate the effectiveness and reliability of the developed armlet with a wet circuit system in accurately measuring the wet circuit in closed channels, an experimental procedure was conducted. The aim of this procedure was to assess the strap's performance under controlled conditions and determine its practical suitability. In this article, we will provide a detailed exploration of the experimental process, highlighting the key steps and considerations involved. The experimental setup commenced with the careful selection of a test channel that closely resembled the dimensions of the target closed channels (Figure S2). The material and geometry of the test channel were meticulously chosen to accurately replicate real-world conditions. Subsequently, the armlet with a wet circuit measurement

system was securely installed within the test channel, following the recommended guidelines provided by the development team. The strap's width, attachment of measurement electrodes, and any necessary fastening mechanisms were implemented in accordance with the design specifications (Figure S3). Prior to proceeding with the experiments, the armlet with a wet circuit measurement system underwent calibration using a reference standard or known flow rates. This calibration step was crucial in ensuring the accuracy and consistency of subsequent measurements.

The experimental procedure involved introducing controlled variations in the flow rate of water or sewage to assess the measurement strap's capability in accurately measuring the wet circuit. These variations simulated different flow conditions typically encountered in closed channels, including low-flow periods, peak flow events, and transitional flow states. Throughout the experiments, continuous data collection took place from the measurement strap's sensors or electrodes. These sensors captured essential parameters such as flow rate, water level, and other variables of interest. The data acquisition system employed was capable of capturing high-resolution measurements at predefined intervals. The collected data from the armlet with wet circuit measurement system were then subjected to comparative analysis, wherein they were compared against reference measurements or established standards. Statistical analyses, including error calculations, and correlation assessments, were conducted to evaluate the strap's accuracy, precision, and reliability in measuring the wet circuit. Based on the comparative analysis, the performance of the measurement strap was comprehensively evaluated, taking into account factors such as measurement accuracy, sensitivity to flow variations, response time, and robustness against environmental factors. These evaluations aimed to determine the strap's suitability for practical use in closed channels (Figure S4). By conducting this rigorous experimental procedure and thoroughly evaluating the performance of the armlet with a wet circuit measurement system, we obtained valuable insights into its effectiveness and reliability in monitoring the wet circuit in closed channels. These findings contribute to advancements in measuring technology, providing more accurate and reliable solutions for monitoring closed channels in real-world scenarios.

### 2.5. Statistical Analysis

The collected data underwent thorough statistical analysis to evaluate the significance of the results and identify potential correlations or patterns. Rigorous testing and performance evaluation were conducted to validate the effectiveness of the novel device. The experimental setup, which included simulated rainwater ingress scenarios, was carefully explained. To assess the device's accuracy, sensitivity, and reliability, specific evaluation metrics were defined. The results obtained from the testing phase were presented and discussed, highlighting the device's precise detection of rainwater ingress. To ensure the accuracy and reliability of the experimental setup and measurements, comprehensive validation procedures were implemented. Quality control measures, including regular instrument calibration and periodic checks, were undertaken to maintain data integrity and minimize potential errors or biases. Data collection procedures were established to capture relevant measurements during the experiments. This involved setting up data loggers or real-time monitoring systems to record sensor readings. The experimental procedures were meticulously designed to simulate various scenarios and conditions encountered in closed channels. Variables such as flow rates, channel dimensions, and moisture levels were carefully controlled. The experiments were conducted systematically, following a predetermined protocol, to ensure consistency and repeatability throughout the study. The pipe slope, inclination, flow of water/sewage, and duration of water/sewage flow (time) were analyzed using Spearman correlation and multiple regression techniques with PAST (PAleontological STatistics), software version 4.23.

## 3. Results

### 3.1. Optimal Armlet Shape for Wet Circuit Measurement System: Key Findings

In this section of the study, a series of laboratory experiments were conducted to create and test various shapes of the armlet with the wet circuit measurement system (Figure 1). The selection process involved examining a range of shapes to determine their impact on wastewater flow in the sewer channel. These shapes included variations in the armlet's casing design. To provide further clarity on our selection, we have included an illustration of the approximate shape of the armlet with the wet circuit system that emerged as the most suitable option after numerous attempts (Figure 2). This shape was chosen based on our experimental data, which demonstrated that it had a minimal impact on wastewater flow when compared to other tested shapes.

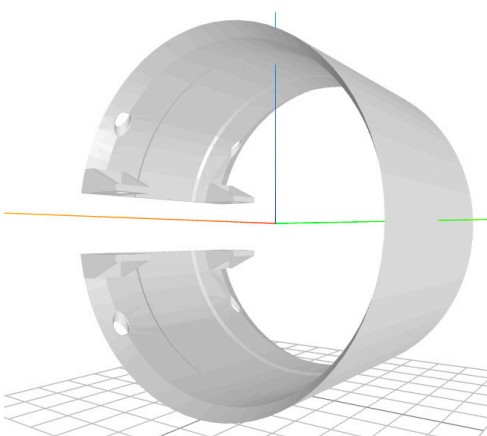

**Figure 1.** 3D visualization of armlet with a wet circuit system, showing a side view. The various colored segments within the figure caption represent the dimensions of an object denoted as 'XYZ,' with each dimension indicated by the colors blue, green, and orange.

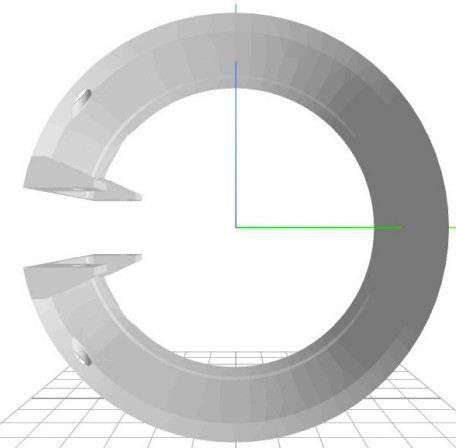

**Figure 2.** Visualization of the measurement armlet with a wet circuit system, showing a front view. The various colored segments within the figure caption represent the dimensions of an object denoted as 'XY,' with each dimension indicated by the colors blue and green.

### 3.2. Results of Laboratory Measurements of Water/Wastewater Flow through the Armlet with Wet Circuit System (Housing of Measurement Straps)

A series of tests were conducted to assess the performance of the measurement strap housing in sewage pipes under various hydraulic loads (Figure 3). Different types of straps made from various materials and in different shapes were tested (Figure 4). The key parameter was whether the tested strap experienced displacement or complete removal by the flowing stream of water/wastewater. Selected results of the measurements regarding

the strap's resistance to displacement under the influence of the flowing water/wastewater are presented in Tables 1–6. Stainless steel plates are affixed to the measurement strap utilizing stainless steel rivets. Two fi8 screws are employed to extend the strap, with each screw exerting a force of 100 N to facilitate its expansion.

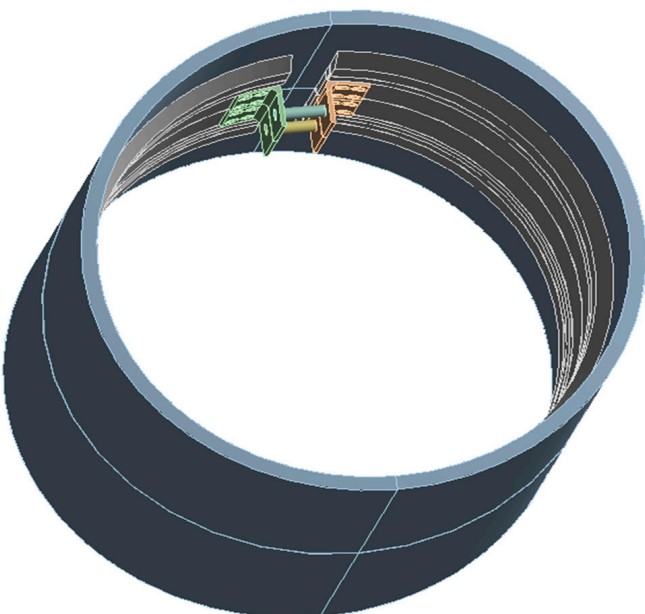

**Figure 3.** Angles joined by rivets in armlet with a wet circuit system.

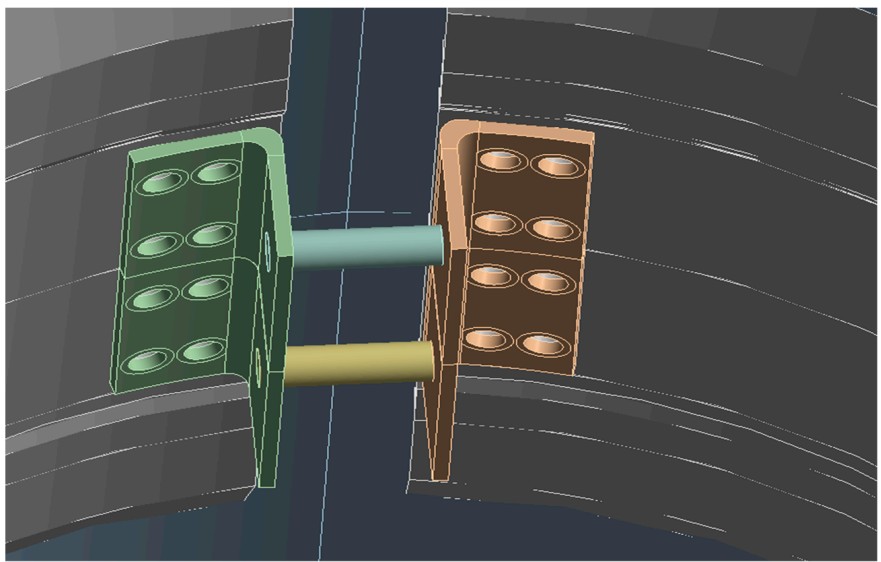

**Figure 4.** Angles joined by rivets.

**Table 1.** Results of sewage flow measurements through the measurement cuff enclosure made of thin-walled PVC (diameter 200 mm).

| Samples | Pipe slope inclination (%) | Water/Sewage Flow (dm³/h) | Splashing of Water/Sewage | Displacement (mm) | Duration of Water/Sewage Flow (h) |
|---|---|---|---|---|---|
| 1 | 0 | 3600 | low | 0 | 0.5 |
| 2 | 5 | 3600 | low | 0 | 0.5 |
| 3 | 10 | 3600 | low | 0 | 0.5 |

**Table 1.** *Cont.*

| Samples | Pipe slope inclination (%) | Water/Sewage Flow (dm³/h) | Splashing of Water/Sewage | Displacement (mm) | Duration of Water/Sewage Flow (h) |
|---|---|---|---|---|---|
| 4 | 20 | 3600 | low | 0 | 0.5 |
| 5 | 50 | 3600 | low | 0 | 0.5 |
| 6 | 100 | 3600 | low | 0 | 0.5 |
| 1 | 0 | 7200 | low | 0 | 5 |
| 2 | 5 | 7200 | low | 0 | 5 |
| 3 | 10 | 7200 | low | 0 | 5 |
| 4 | 20 | 7200 | low | 0 | 5 |
| 5 | 50 | 7200 | low | 0 | 5 |
| 6 | 100 | 7200 | low | 0 | 5 |

**Table 2.** Results of sewage flow measurements through the measurement cuff enclosure made of ABS ULTRAT (diameter 160 mm).

| Samples | Pipe Slope Inclination (%) | Water/Sewage Flow (dm³/h) | Splashing of Water/Sewage | Displacement (mm) | Duration of Water/Sewage Flow (h) |
|---|---|---|---|---|---|
| 1 | 0 | 3600 | low | 0 | 0.5 |
| 2 | 5 | 3600 | low | 0 | 0.5 |
| 3 | 10 | 3600 | low | 0 | 0.5 |
| 4 | 20 | 3600 | low | 0 | 0.5 |
| 5 | 50 | 3600 | low | 0 | 0.5 |
| 6 | 100 | 3600 | low | 0 | 0.5 |
| 1 | 0 | 7200 | low | 0 | 5 |
| 2 | 5 | 7200 | low | 0 | 5 |
| 3 | 10 | 7200 | low | 0 | 5 |
| 4 | 20 | 7200 | low | 0 | 5 |
| 5 | 50 | 7200 | low | 0 | 5 |
| 6 | 100 | 7200 | low | 0 | 5 |

**Table 3.** Results of sewage flow measurements through the measurement cuff enclosure (thin-walled PCV, outer diameter 315 mm, inner diameter 289 mm).

| Samples | Pipe Slope Inclination (%) | Water/Sewage Flow (dm³/h) | Splashing of Water/Sewage | Displacement (mm) | Duration of Water/Sewage Flow (h) |
|---|---|---|---|---|---|
| 1 | 0 | 3600 | low | 0 | 0.5 |
| 2 | 5 | 3600 | low | 0 | 0.5 |
| 3 | 10 | 3600 | low | 0 | 0.5 |
| 4 | 20 | 3600 | low | 0 | 0.5 |
| 5 | 50 | 3600 | low | 0 | 0.5 |
| 6 | 100 | 3600 | low | 0 | 0.5 |
| 1 | 0 | 7200 | low | 0 | 5 |
| 2 | 5 | 7200 | low | 0 | 5 |
| 3 | 10 | 7200 | low | 0 | 5 |
| 4 | 20 | 7200 | low | 0 | 5 |
| 5 | 50 | 7200 | low | 0 | 5 |
| 6 | 100 | 7200 | low | 0 | 5 |

**Table 4.** Results of sewage flow measurements through the measurement cuff enclosure (ASA material, outer diameter 200 mm).

| Samples | Pipe Slope Inclination (%) | Water/Sewage Flow (dm$^3$/h) | Splashing of Water/Sewage | Displacement (mm) | Duration of Water/Sewage Flow (h) |
|---|---|---|---|---|---|
| 1 | 0 | 3600 | low | 0 | 0.5 |
| 2 | 5 | 3600 | low | 0 | 0.5 |
| 3 | 10 | 3600 | low | 0 | 0.5 |
| 4 | 20 | 3600 | low | 0 | 0.5 |
| 5 | 50 | 3600 | low | 0 | 0.5 |
| 6 | 100 | 3600 | low | 0 | 0.5 |
| 1 | 0 | 7200 | low | 0 | 5 |
| 2 | 5 | 7200 | low | 0 | 5 |
| 3 | 10 | 7200 | low | 0 | 5 |
| 4 | 20 | 7200 | low | 0 | 5 |
| 5 | 50 | 7200 | low | 0 | 5 |
| 6 | 100 | 7200 | low | 0 | 5 |

**Table 5.** Results of sewage flow measurements through the measurement cuff enclosure (black ABS material, outer diameter 160 mm, non-beveled wall shape).

| Samples | Pipe Slope Inclination(%) | Water/Sewage Flow (dm$^3$/h) | Splashing of Water/Sewage | Displacement (mm) | Duration of Water/Sewage Flow (h) |
|---|---|---|---|---|---|
| 1 | 0 | 3600 | medium | 0 | 0.5 |
| 2 | 5 | 3600 | medium | 0 | 0.5 |
| 3 | 10 | 3600 | medium | 0 | 0.5 |
| 4 | 20 | 3600 | medium | 0 | 0.5 |
| 5 | 50 | 3600 | medium | 0 | 0.5 |
| 6 | 100 | 3600 | medium | 0 | 0.5 |
| 1 | 0 | 7200 | medium | 0 | 5 |
| 2 | 5 | 7200 | medium | 0 | 5 |
| 3 | 10 | 7200 | medium | 0 | 5 |
| 4 | 20 | 7200 | medium | 0 | 5 |
| 5 | 50 | 7200 | medium | 0 | 5 |
| 6 | 100 | 7200 | medium | 0 | 5 |

**Table 6.** Results of sewage flow measurements through measurement cuff enclosure number 3 (outer diameter 250 mm, inner diameter 230 mm).

| Samples | Pipe Slope Inclination (%) | Water/Sewage Flow (dm$^3$/h) | Splashing of Water/Sewage | Displacement (mm) | Duration of Water/Sewage Flow (h) |
|---|---|---|---|---|---|
| 1 | 0 | 3600 | low | 0 | 0.5 |
| 2 | 5 | 3600 | low | 0 | 0.5 |
| 3 | 10 | 3600 | low | 0 | 0.5 |
| 4 | 20 | 3600 | low | 0 | 0.5 |
| 5 | 50 | 3600 | low | 0 | 0.5 |
| 6 | 100 | 3600 | low | 0 | 0.5 |
| 1 | 0 | 7200 | low | 0 | 5 |
| 2 | 5 | 7200 | low | 0 | 5 |
| 3 | 10 | 7200 | low | 0 | 5 |
| 4 | 20 | 7200 | low | 0 | 5 |
| 5 | 50 | 7200 | low | 0 | 5 |
| 6 | 100 | 7200 | low | 0 | 5 |

The pressure at the point where the pipe and the measurement strap come into contact exceeded 5000 Pa. The force applied at the interface between the strap and the pipe was measured to be 1138.4 N. The proposed solutions for expanding the strap and implementing the clamping mechanism were illustrated for installation on a pipe with a diameter of 250 mm. Specifically, a Roman screw featuring opposing threads was conceptualized, fabricated, and subjected to testing (photographs are available for reference). The maximal deformations recorded were within the range of 0.2 mm. Further investigation revealed that the peak stress experienced by the strap was 4.46 MPa. When the pipe diameter was 160 mm, the stress level reached its maximum at 1.49 MPa (Figure 5). For a pipe with a diameter of 200 mm, the stress level measured 0.8 MPa, as illustrated in Figure 6. In the case of a 250 mm diameter pipe, the maximum stress level was recorded at 0.65 MPa, as depicted in Figure 7. When dealing with a 315 mm diameter, the maximum stress level dropped to 0.51 MPa (Figure 8). Finally, for a 400 mm diameter pipe, the maximum stress level was at its lowest, measuring 0.41 MPa, as indicated in Figure 9.

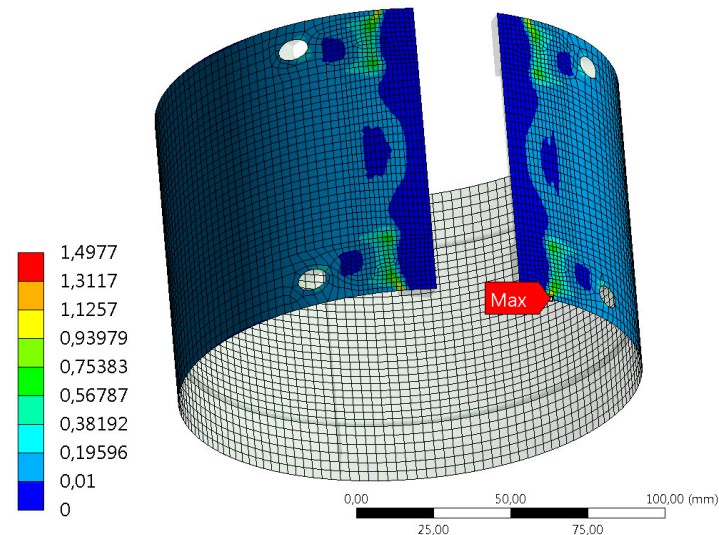

**Figure 5.** Visualizing pressure distribution (MPa) in armlet with a wet circuit measurement system for a 160 mm diameter pipe.

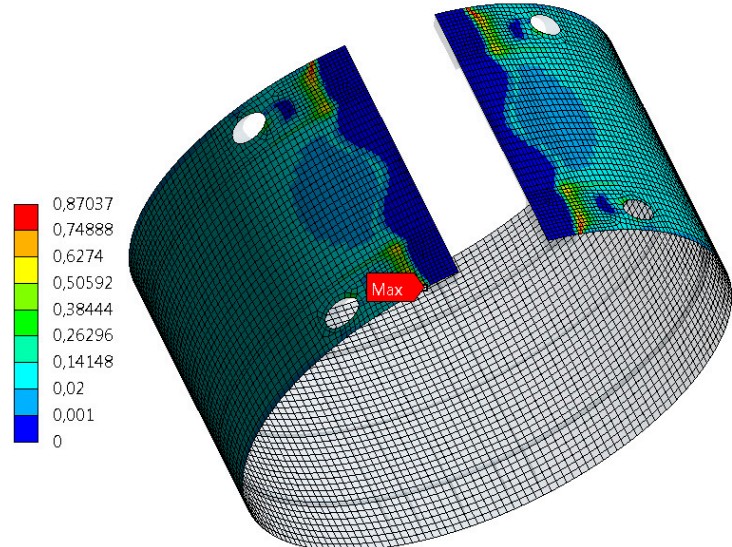

**Figure 6.** Visualizing pressure distribution (MPa) in armlet with a wet circuit measurement system for a 200 mm diameter pipe.

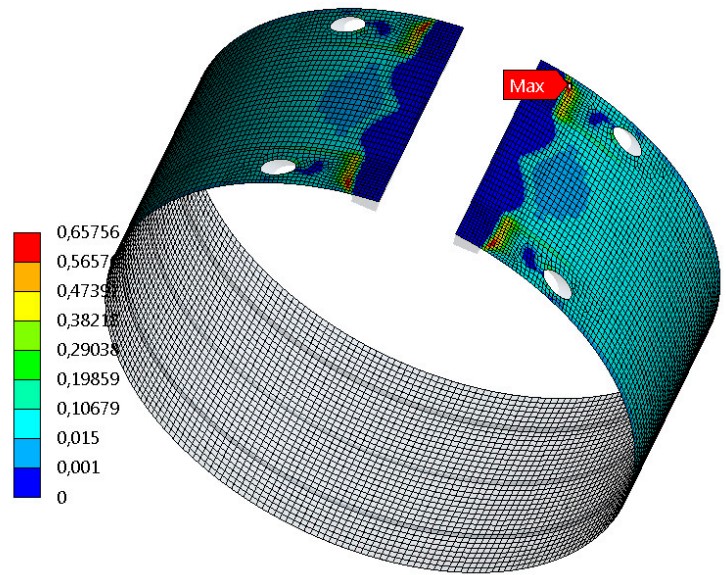

**Figure 7.** Visualizing pressure distribution (MPa) in armlet with a wet circuit measurement system for a 250 mm diameter pipe.

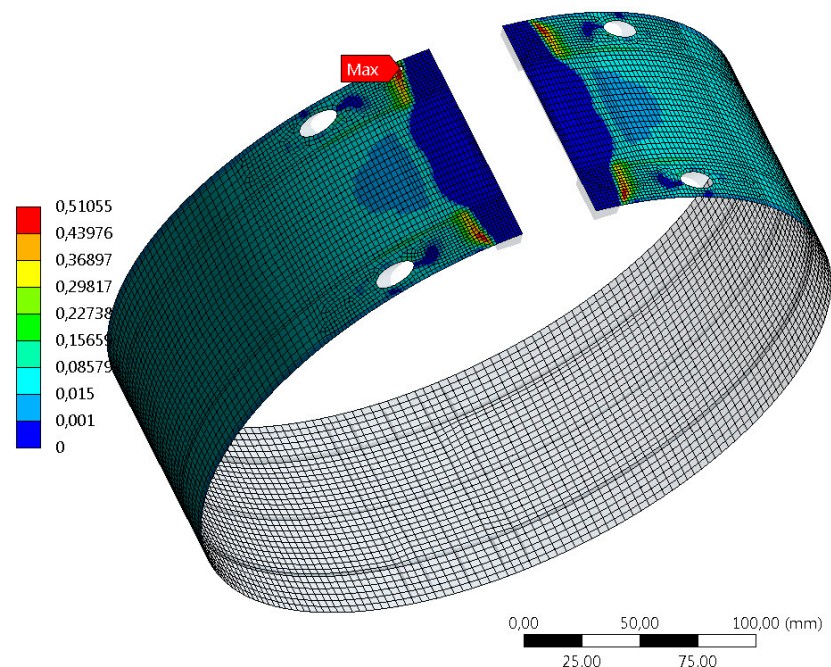

**Figure 8.** Visualizing pressure distribution (MPa) in armlet with a wet circuit measurement system for a 315 mm diameter pipe.

Upon thorough analysis, it was concluded that the most suitable material should possess superior yield strength and optimal tensile strength at the point of fracture. Consequently, the recommended material for this application was identified as PC/ABS. Furthermore, Spearman correlation analyses were conducted to assess the relationship between different parameters of the selected PC/ABS material (Figure 10).

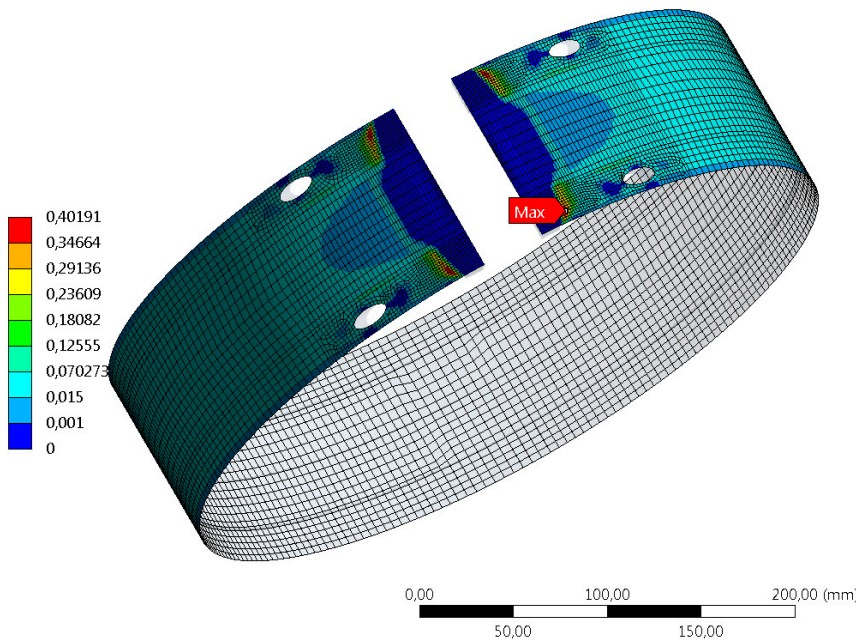

**Figure 9.** Visualizing pressure distribution (MPa) in armlet with a wet circuit measurement system for a 400 mm diameter pipe.

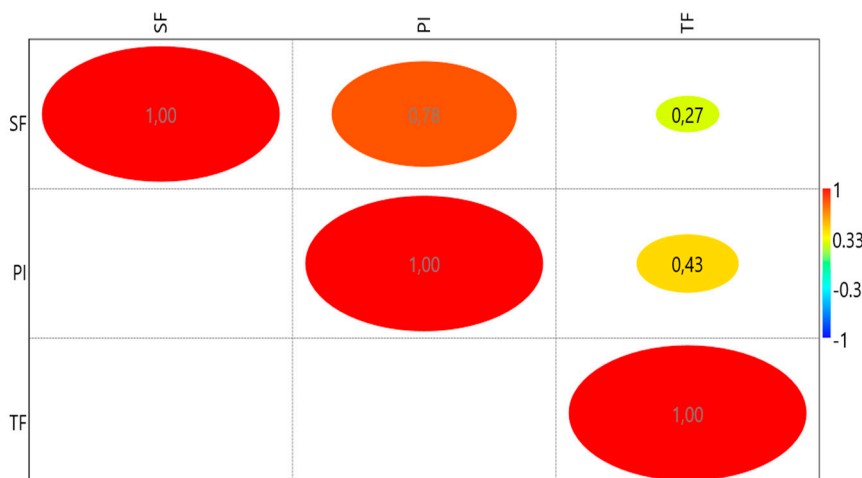

**Figure 10.** Spearman correlation for parameters used in PC/ABS. SF—sewage flow; PI—pipe slope inclination; TF—time of water/sewage flow.

## 4. Discussion

### 4.1. Rainwater Ingress Detection Device for Enhanced Wastewater Management

Accurate detection and quantification of rainwater ingress in sewer channels are crucial for effective wastewater management. The novel rainwater ingress detection device utilizes advanced sensor technology to measure parameters such as water level, flow rate, and water quality in real-time [34]. By analyzing the data, the device accurately detects rainwater ingress events and distinguishes them from normal wastewater flow using sophisticated algorithms. It also quantifies the volume and frequency of rainwater infiltration, providing valuable information for assessing the sewer system's capacity and making informed decisions regarding maintenance, upgrades, and resource allocation. The trend indicates that as the pipe diameter increases, the stress level decreases. This inverse relationship is particularly evident when comparing the stress levels at 160 mm and 400 mm diameters, where the stress decreases from 1.49 MPa (Figure 5) to 0.41 MPa (Figure 9). The monitoring of wet circuits in closed channels and examining the connections between

key parameters, including sewage flow (SF), pipe slope (PI), and time of water/sewage flow (TF), holds significant importance (Figure 10). These factors are pivotal for efficient water and sewage system management and design. Sewage flow (SF) represents the sewage or wastewater flow rate, which is crucial for sizing pipes to handle the generated sewage volume. Pipe slope (PI), indicating pipe inclination, impacts flow velocity and efficiency, with steeper slopes allowing faster flow but potentially requiring more energy for pumping. Time of water/sewage flow (TF) refers to flow duration, which is vital for resource management and insights into system operation. Interpreting the results provides guidance for system adjustments, design choices, and resource allocation in innovations for urban wastewater management and advancing sewage transformation strategies.

The other results demonstrated bi-LSTM's (bidirectional Long Short-Term Memory) effectiveness in predicting time series, particularly in forecasting wastewater flow for a treatment plant. The approach successfully achieved sequence-to-point p-step prediction by using past information and a sliding time window. This highlights the potential of deep neural networks for accurate forecasting in practical applications like wastewater management. Further research should explore scalability and compare bi-LSTM with other methods for enhanced efficiency in diverse fields [35]. The hydraulic performance of all tested measurement straps was generally positive. The tests were conducted at flow rates of 3600 and 7200 $dm^3$/h. A short trial of up to 0.5 h and a long trial of 5 h were carried out separately for each flow rate. Notably, none of the tested measurement straps exhibited any displacement.

Rainwater ingress in sewer channels poses significant challenges for urban wastewater management systems, leading to capacity issues, increased treatment costs, and potential environmental pollution. This paper presents the development of a novel device aimed at enhancing rainwater ingress detection in sewer channels. The device provides real-time monitoring capabilities and enables proactive maintenance strategies, thereby improving the overall performance and resilience of urban wastewater management systems [36]. This paper delves into the meticulous design, implementation, and assessment of a groundbreaking device with the utmost potential to optimize resource allocation and effectively counteract the detrimental effects of rainwater infiltration. Accurate measurement of the wet circuit within enclosed channels, such as sewage or stormwater channels, holds paramount importance for the efficient management and maintenance of wastewater systems. In this article, we proudly unveil our innovative creation—an armlet with a wet circuit measurement system explicitly tailored to deliver precise and dependable measurements in these channels (Table 1). Splashing was minimal and primarily dependent on the shape of the strap, with the proper shaping of the front wall being crucial (Table 2). This allowed water to flow through the strap smoothly, minimizing turbulence. However, it should be noted that the measurement strap made of black ABS, which lacked edge shaping, resulted in noticeable water splashing and dispersion. Through the analysis process, a final effective shape for the measurement strap was developed and implemented.

The implementation of the rainwater ingress detection device in wastewater management systems has several practical implications. Firstly, it facilitates proactive maintenance strategies by identifying areas with high rainwater ingress [37]. Maintenance crews can prioritize these areas for inspections and repairs, preventing potential infrastructure damage and reducing the risk of sanitary sewer overflows during heavy rainfall events. Secondly, the device's data can be integrated into the existing wastewater management system, providing valuable insights for decision-making processes [38]. The accurate quantification of rainwater ingress allows for optimized resource allocation, cost-effective infrastructure planning, and the development of targeted strategies to manage rainwater infiltration. It is worth noting that the effectiveness of the rainwater ingress detection device may vary depending on factors such as the sewer network's complexity, the density of monitoring points, and the reliability of the data transmission and analysis systems. Further research and development are required to refine the device's performance and address any potential limitations.

### 4.2. Advancing Measurement Technology for Monitoring Wet Circuits in Closed Channels

To ensure accurate results, it is imperative to have a properly sized strap. Our recommended width of 100 mm allows for secure attachment of the measurement electrodes, minimizing any potential disruptions during the measurement process. A key feature of our measurement strap is the incorporation of a phase on the side facing the flowing water or sewage. This ingenious design element facilitates a seamless flow, effectively preventing any disturbances or interferences that might compromise the accuracy of the measurements. Through careful consideration of the hydraulic dynamics within the channel, the strap guarantees reliable and consistent data collection (Table 3). Efficient installation is essential for widespread adoption and practical implementation of our measurement strap. To address this, we propose the use of a Roman screw with opposing threads. This innovative fastening mechanism streamlines the installation process, eliminating the need for additional tools. Alternatively, an expanding mechanism utilizing a fixed regular screw or a silicone cone inserted between the angle brackets of the thin-walled PVC strap offers easy and secure installation options.

Incorporating special openings in the strap design further facilitates the installation of the expanding mechanism for the measurement strap housing. These purpose-built features enhance the usability and functionality of the strap, ensuring seamless integration into existing channel infrastructures. The unique shape of our measurement strap allows for collision-free mounting of the measurement electrodes. Applied on a flexible PCB, the electrodes are meticulously positioned to capture precise measurements without interference. To ensure their secure placement, we recommend utilizing additional straps with snaps for proper fixation. This robust fixation system guarantees the electrodes remain firmly in place throughout the monitoring process. Rainwater ingress in sewer channels is a persistent challenge faced by urban wastewater management systems worldwide [39]. The infiltration of rainwater into sewer networks not only imposes significant strain on treatment plants but also raises concerns regarding environmental contamination and system capacity overload. Detecting and addressing rainwater ingress in a timely manner is crucial to ensure the optimal functioning and resilience of urban wastewater management systems [40]. In addition to rainwater ingress detection, the device enables proactive maintenance strategies to mitigate the adverse impacts of infiltration. Early detection of rainwater ingress allows wastewater management authorities to implement immediate corrective actions, such as targeted cleaning, optimized pump operations, and diversion of excess flows to alternative treatment facilities. These proactive measures contribute to improving system efficiency, reducing treatment costs, and minimizing the risk of overflows and environmental contamination.

### 4.3. Innovations for Urban Wastewater Management: Advancing Sewage Transformation Strategies

The measurement strap's successful development marks a major advancement in monitoring wet circuits in closed channels. With precise and reliable measurements, our strap empowers wastewater management professionals to make informed decisions on maintenance, resource allocation, and system optimization. This proactive approach mitigates risks like flooding and environmental pollution caused by rainwater infiltration. The device accurately detects and quantifies rainwater ingress in sewer channels, enabling real-time monitoring, data analysis, and effective strategies to minimize its impact [41]. The precision, reliability, and practicality of our measurement strap offer tangible solutions for accurate measurements in closed channels [42,43]. By embracing these innovations, municipalities and wastewater management authorities can enhance their understanding of wet circuits, make data-driven decisions, and safeguard the environment against the perils of rainwater infiltration. Efficient rainwater management is vital for urban wastewater systems facing flood and contamination risks. Our innovative armlet with a wet circuit measurement system is tailored for enclosed conduits like sewage and stormwater channels. Precision engineering replicates closed channel conditions, aided by advanced materials

and sensors for real-time rainwater detection. This data-driven approach, rooted in rigorous lab experiments, explores various armlet shapes and sizes to minimize interference with wastewater flow. Accurate detection and quantification of rainwater enable proactive maintenance, averting floods and environmental harm. This innovative solution extends beyond monitoring, addressing rainwater-related challenges, enhancing urban wastewater management efficiency and sustainability.

While this research represents a significant advancement in rainwater ingress detection, further studies are needed to address certain challenges [34,37,44]. Long-term durability and reliability of the device, seamless integration with existing management systems, and scalability for large-scale implementation are areas that require continued investigation [45]. Future research may explore the device with predictive analytics and machine learning for accurate forecasting and intelligent decision-making. The soft sensor accurately estimated wastewater flowrates, closely matching manual measurements (207 and 205 $m^3$/h vs. 212 and 207 $m^3$/h) and comparing well to reference flowrates (approximately 188 and 180 $m^3$/h). This demonstrates its precision, particularly in large sewer lines (diameter 1 m) when the distance sensor operated flawlessly, enabling precise flowrate estimation [46]. Moreover, the device's compatibility with existing sewer infrastructure and easy installation were confirmed during field tests. It seamlessly integrated into the sewer system without requiring extensive modifications or disruptions to normal operations [47]. The non-intrusive measurement techniques employed by the device also ensured minimal impact on the system's hydraulic performance and overall functionality [48]. The newly developed device brings forth numerous benefits for urban wastewater management systems [49]. Its ability to promptly identify rainwater infiltration events enables targeted cleaning, optimized pump operations, and diversion of excess flows, thereby reducing the risk of system overload, backups, and environmental contamination.

## 5. Conclusions

We have devised a measurement strap, purpose-built to meticulously gauge wet circuits within enclosed conduits, such as sewage or stormwater channels. For trustworthy outcomes, it is imperative that the strap maintains a width of 100 mm, facilitating secure attachment of measurement electrodes. The strategic incorporation of a phase on the side facing the flowing water or sewage stream within the strap's design fosters an uninterrupted flow. For seamless installation without the need for supplementary tools, the recommendation is to employ a Roman screw with opposing threads to firmly secure the strap within the channel. Alternatively, an expanding mechanism utilizing a fixed regular screw or a silicone cone wedged between the angle brackets of the thin-walled PVC strap can be explored for effortless setup. To streamline the installation of the expanding mechanism within the measurement strap housing, it is vital to integrate the dedicated openings into the strap's design. The unique configuration of the measurement strap allows collision-free positioning of the measurement electrodes on a flexible PCB. Furthermore, the use of additional straps with snaps guarantees proper fixation of the measurement electrodes. Our conducted analysis has underscored the optimal selection of a material with the highest yield strength and supreme tensile strength at the point of rupture. Consequently, the endorsed material is PC/ABS. The application of these recommendations culminates in the creation of a measurement strap that yields precise and dependable measurements of wet circuits across diverse channel applications. Beyond this, the invention of a groundbreaking rainwater ingress detection apparatus bears significant potential for enhancing urban wastewater management. By accurately identifying and quantifying rainwater ingress within sewer channels, this device empowers proactive maintenance, resource allocation optimization, and heightened system efficiency. Its real-time monitoring, data analysis, and quantification capabilities form the bedrock for well-informed decision-making, effectively mitigating the perils of flooding and environmental contamination arising from rainwater infiltration.

**Supplementary Materials:** The following supporting information can be downloaded at: https://www.mdpi.com/article/10.3390/app131910892/s1, Figure S1: BS ULTRAT measurement strap securely attached in a streamlined shape, printed using 3D technology; Figure S2: View of the research setup during hydraulic load testing with water flow; Figure S3: View of three example straps: on the left, a strap made of ABS ULTRAT; in the middle, a wider strap made of brown ABS; on the right, a strap made of a PVC flat bar; Figure S4: Examples of mounted straps on a 400 mm diameter sewage pipe: in the middle, a wide strap made of brown ABS; on the left, a narrow strap made of black ABS.

**Author Contributions:** Conceptualization, T.S. and K.C.; methodology, T.S. and K.C.; software, W.H.; validation, W.H.; writing—original draft preparation, T.S.; writing—review and editing, W.H.; visualization, T.S. and W.H.; supervision, T.S. and K.C. All authors have read and agreed to the published version of the manuscript.

**Funding:** This research received no external funding.

**Institutional Review Board Statement:** Not applicable.

**Informed Consent Statement:** Not applicable.

**Data Availability Statement:** Data are contained within the article or Supplementary Materials.

**Acknowledgments:** This publication received funding from the Ministry of Education and Science for University of Science and Technology in 2023.

**Conflicts of Interest:** The authors declare no conflict of interest.

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
