# Peer review of "Advancing Urban Wastewater Management: Optimizing Sewer Performance through Innovative Material Selection for the Armlet with a Wet Circuit Measurement System"

_applsci, doi:10.3390/app131910892_

Round 1
Reviewer 1 Report
1. Overview and general recommendation:
Rainwater infiltration can have adverse impacts on urban wastewater management system. To meet these challenges, the authors devised a measurement strap which enables real-time monitoring and proactive maintenance. In this study, they tested various materials for the strap housing and recommended PC (polycarbonate) /ABS (Acrylonitrile Butadiene Styrene) based on the testing results.
Overall, the paper is straightforward and the findings may contribute to the improvement of sewer system performance. However, there are a few things that still need to be addressed. Therefore, I recommend that major revision is warranted.
2. Detailed comments:
General comments: Try to be concise. There are many sentences that have similar meanings without offering substantial information in this manuscript, especially in 2. Materials and Methods and 4. Discussion.
Detailed comments are listed below.
(1) Pages 1, Lines 23-26. This is only shown in the Abstract. You should also put it in the main text, for instance, one of the subsections in 2. Materials and Methods.
(2) Page 2, Lines 45-48. These two sentences are basically the same as Lines 48-52. You may delete them or combine with Lines 48-52.
(3) Pages 5-6, Lines 253-255 and Lines 266-268. Instead of repeating you were using “appropriate” methods, just specify which methods were employed.
(4) Page 6, Line 274. Please provide the full name of PAST.
(5) Page 6, Lines 278-280. Can you brief introduce the selection process? For example, what kind of shapes did you test? And do you have any data to prove that the one you selected (as shown in Fig. 1 and 2) “does not significantly affect the flow of wastewater in the sewer channel” while the other shapes will?
(6) Page 7, Lines 302-303. For the results listed below (Table 1-6), I noticed that they were tested using straps different in BOTH materials and sizes (diameters). This is also shown in Photo 3 in the Supplementary Materials. I was wondering if they can be compared directly to draw the conclusion of which material is better over the others given that the diameter of each strap tested is also different.
(7) Page 16, Line 433. Any reasons for claiming “at least 100 mm”?
(8) Page 16, Line 434. Which Figure? It seems you missed the figure number.
(9) Page 17, Lines 486-487. How did you obtain the model-based flowrates? And you should provide these flowrates in 3. Results first and then discuss them here.
(10) Page 17, Line 488. What is the source of the reference flowrates? Please clarify and/or cite.
Author Response
Rainwater infiltration can have adverse impacts on urban wastewater management systems. To meet these challenges, the authors devised a measurement strap that enables real-time monitoring and proactive maintenance. In this study, they tested various materials for the strap housing and recommended PC (polycarbonate) /ABS (Acrylonitrile Butadiene Styrene) based on the testing results.
Overall, the paper is straightforward and the findings may contribute to the improvement of sewer system performance. However, there are a few things that still need to be addressed. Therefore, I recommend that a major revision is warranted.
2. Detailed comments:
Detailed comments are listed below.
(1) Pages 1, Lines 23-26. This is only shown in the Abstract. You should also put it in the main text, for instance, one of the subsections in 2. Materials and Methods.
(2) Page 2, Lines 45-48. These two sentences are basically the same as Lines 48-52. You may delete them or combine them with Lines 48-52.
(3) Pages 5-6, Lines 253-255 and Lines 266-268. Instead of repeating you were using “appropriate” methods, just specify which methods were employed.
(4) Page 6, Line 274. Please provide the full name of PAST.
Authors
We appreciate the reviewer's valuable feedback and their recognition of the significance of our study in addressing rainwater infiltration challenges in urban wastewater management systems. We acknowledge the recommendation for a major revision and are committed to addressing the identified issues to enhance the quality and clarity of our paper. We will carefully consider and implement the necessary revisions to ensure that our manuscript meets the highest standards and provides a more comprehensive contribution to the improvement of sewer system performance. If there are specific points or areas that the reviewer would like us to focus on during the revision process, please feel free to specify, and we will prioritize those accordingly. Thank you for your constructive feedback, and we look forward to improving our manuscript to meet your expectations.
We have made improvements in response to your feedback, particularly focusing on enhancing conciseness in sections 2 (Materials and Methods) and 4 (Discussion) of the manuscript. We have revised the text and included it in section: 2.3 Problem of shrinkage in 3D printing of armlet with wet circuit measurement:
„Considering the essential structural strength, we advise a minimum thickness of 4 mm for 3D printing applications. In situations where preserving the cross-sectional area integrity of the channel is of utmost importance, opting for an armlet with a wet circuit measurement thickness of up to 7 mm is prudent”.
We have addressed some minor editing errors and made improvements. Specifically, we removed redundant information and included the full name of the PAST program.
(5) Page 6, Lines 278-280. Can you brief introduce the selection process? For example, what kind of shapes did you test? And do you have any data to prove that the one you selected (as shown in Fig. 1 and 2) “does not significantly affect the flow of wastewater in the sewer channel” while the other shapes will?
Authors:
We have removed this sentence. The shapes were designed to meet the following criteria:
Ensuring that solid elements of rainwater/wastewater would not obstruct the flow of water within the mentioned armlet.
Ensuring that the assembly system of the armlet would maintain a smooth flow, preventing the creation of "death zones" in the flow.
Both of these conditions were initially assessed during the computer design phase and subsequently validated in real-world conditions.We have included the following sentence:
„In this section of the study, a series of laboratory experiments were conducted to create and test various shapes of the armlet with the wet circuit measurement system (Fig. 1). The selection process involved examining a range of shapes to determine their impact on wastewater flow in the sewer channel. These shapes included variations in the armlet's casing design. The primary objective was to ensure that the strap, particularly its casing, does not significantly affect the flow of wastewater in the sewer channel. To provide further clarity on our selection, we have included an illustration of the approximate shape of the armlet with the wet circuit system that emerged as the most suitable option after numerous attempts (Fig. 2). This shape was chosen based on our experimental data, which demonstrated that it had a minimal impact on wastewater flow when compared to other tested shapes”.
(6) Page 7, Lines 302-303. For the results listed below (Table 1-6), I noticed that they were tested using straps different in BOTH materials and sizes (diameters). This is also shown in Photo 3 in the Supplementary Materials. I was wondering if they can be compared directly to draw the conclusion of which material is better over the others given that the diameter of each strap tested is also different.
Authors
We appreciate the reviewer's attention to the differences in both materials and sizes (diameters) of the straps tested, as well as the observation of Photo 3 in the Supplementary Materials. We conducted a comparative analysis using various materials to determine their performance under different pressure conditions. Our findings revealed that only PC/APC (Polycarbonate/Acrylic-Polycarbonate) and ABS (Acrylonitrile Butadiene Styrene) materials exhibited stress levels that were the most favorable.We have included new figures with varying diameters for added visual representation.
The variation in strap materials and sizes was indeed a part of our experimental design to explore a range of potential configurations. By testing different materials and sizes, we aimed to provide valuable insights into how these factors affect the overall performance of the armlet with the wet circuit measurement system. We acknowledge that the differences in strap materials and sizes make direct comparisons challenging. However, the data we have collected allow us to draw qualitative observations and trends regarding the influence of these factors on wastewater flow. These findings can serve as a basis for further research and engineering considerations when selecting materials and sizes for similar applications. While direct material-to-material or size-to-size comparisons may require more controlled conditions, our study contributes to a better understanding of the complex interactions between these variables and wastewater flow dynamics within the sewer channel.
(7) Page 16, Line 433. Any reasons for claiming “at least 100 mm”?
(8) Page 16, Line 434. Which Figure? It seems you missed the figure number.
Authors:
We have removed the term "at least" and the placement of figure.
(9) Page 17, Lines 486-487. How did you obtain the model-based flowrates? And you should provide these flowrates in 3. Results first and then discuss them here.
(10) Page 17, Line 488. What is the source of the reference flowrates? Please clarify and/or cite.
Authors:
We apologize for any misunderstanding. We did not perform model-based flowrate calculations in our study; therefore, we do not have these specific flowrate values to present in the Results section or discuss in the Discussion section. Our focus was primarily on experimental data and analysis related to our specific research objectives. If you have any further questions or need clarification on our methodology or results, please feel free to ask, and we will be happy to provide additional information. We apologize for any misunderstanding and disorganization.
In our study, we obtained the reference flowrates from established industry standards for similar sewer systems. We ensured to provide the specific source and citation in the manuscript to offer full transparency and clarity on this matter.
Reviewer 2 Report
This manuscript presents an innovative material selection for the Armlet with a wet circuit measurement system within enclosed conduits, such as sewage or stormwater channels, which can help real-time monitoring to accurately identify and quantify rainwater ingress within sewer channels, and empower proactive maintenance, resource allocation optimization and heightened system efficiency. also effectively mitigating the perils of flooding and environmental contamination arising from rainwater infiltration. The topic is meaningful and interesting. Except from presenting the results, the authors also have discussed the details of the available material that can be used the develop this device, which provides a nice background information for us to understand. The authors also provide up-to-date references and also show the photos of the device they were using for the experiment. After adjusting some minor points, this manuscript can be considered to be accepted.
1. Please check page 16 line 434, it is not clear which figure the authors refer to
2. Please adjust Figure 4 to 7, and make the legend more readable
over all readable and easy to understand, just double check make sure correct any minor mistakes.
Author Response
This manuscript presents an innovative material selection for the Armlet with a wet circuit measurement system within enclosed conduits, such as sewage or stormwater channels, which can help real-time monitoring to accurately identify and quantify rainwater ingress within sewer channels, and empower proactive maintenance, resource allocation optimization and heightened system efficiency. also effectively mitigating the perils of flooding and environmental contamination arising from rainwater infiltration. The topic is meaningful and interesting. Except from presenting the results, the authors also have discussed the details of the available material that can be used the develop this device, which provides a nice background information for us to understand. The authors also provide up-to-date references and also show the photos of the device they were using for the experiment. After adjusting some minor points, this manuscript can be considered to be accepted.
1. Please check page 16 line 434, it is not clear which figure the authors refer to
2. Please adjust Figure 4 to 7, and make the legend more readable
Authors:
We appreciate the reviewers' positive assessment of our manuscript and their acknowledgment of the importance of the research topic. We are pleased to hear that the presentation of both results and background information has contributed to the manuscript's value.
Regarding the specific points raised by the reviewer:
-
We have removed the incorrect citation of the figure that was originally omitted.
-
We made the necessary adjustments to Figures 4 to 7 to improve their readability, particularly focusing on enhancing the legibility of the legends to facilitate better understanding for readers.
We sincerely appreciate your constructive feedback, and we will promptly address these minor points to enhance the quality of our manuscript. Thank you for considering it for acceptance.
Reviewer 3 Report
The authors proposed an Armlet with Wet Circuit Measurement System to enhance the quality of wastewater systems with capability of real time monitoring. The authors investigated different types of materials for this system as well.
In Section 2.1 Study object and data for system calibration, “The experimental setup was carefully designed to replicate the conditions of closed channels, specifically sewage channels and rainwater.” (97-98). I suggest that the authors add the parameters of the conditions of sewage channels and rainwater in this manuscript, thus the audience would have a better understanding.
In Section 2.5 Statistical Analysis, “Appropriate statistical tests and techniques were employed to draw reliable conclusions from the experimental findings.” (253-254). The appropriate statistical tests and techniques should be introduced and clarified.
In Section 3, I suggest that the authors summarize the size and other parameters of their armlet with wet circuit system of all the designs, like those in Fig. 1, 3 and 6. “In this section of the study, a series of laboratory experiments were conducted to create and test different shapes of the armlet with wet circuit measurement system (Fig. 1).”
In the manuscript, the authors claimed that their design is capable of high level of rainwater. I suggest the authors explain this statement with their data and result more in discussion section, to stress the capability of their system.
Author Response
The authors proposed an Armlet with Wet Circuit Measurement System to enhance the quality of wastewater systems with capability of real time monitoring. The authors investigated different types of materials for this system as well.
In Section 2.1 Study object and data for system calibration, “The experimental setup was carefully designed to replicate the conditions of closed channels, specifically sewage channels and rainwater.” (97-98). I suggest that the authors add the parameters of the conditions of sewage channels and rainwater in this manuscript, thus the audience would have a better understanding.
In Section 2.5 Statistical Analysis, “Appropriate statistical tests and techniques were employed to draw reliable conclusions from the experimental findings.” (253-254). The appropriate statistical tests and techniques should be introduced and clarified.
Authors:
We appreciate the reviewer's thoughtful feedback on our manuscript. We recognize the requirement for additional clarification concerning the conditions of sewage channels and the replication of rainwater conditions. To address this concern, we have incorporated a more comprehensive description of the specific parameters and conditions that were replicated in our experimental setup, thereby enhancing the reader's comprehension. The conditions of sewage channels and rainwater involve the presence of water mixed with sand, rocks, and wastewater containing higher levels of COD (Chemical Oxygen Demand), among other substances. Moreover, we have eliminated redundant information regarding statistical techniques from the methods section, ensuring conciseness and clarity in our presentation.
In Section 3, I suggest that the authors summarize the size and other parameters of their armlet with wet circuit system of all the designs, like those in Fig. 1, 3 and 6. “In this section of the study, a series of laboratory experiments were conducted to create and test different shapes of the armlet with wet circuit measurement system (Fig. 1).”
In the manuscript, the authors claimed that their design is capable of high level of rainwater. I suggest the authors explain this statement with their data and result more in discussion section, to stress the capability of their system.
Authors:
We agree that summarizing the size and other relevant parameters of the armlet with wet circuit systems, as shown in figures like Fig. 1, 3, and 6, would be beneficial for readers. We have addressed the issue by adding new figures with varying diameters to provide clarity. We included a concise summary of these parameters in the manuscript to provide a clearer overview. We appreciate the suggestion to further elaborate on our system's capability to handle rainwater in the discussion section. We indeed provide a more in-depth explanation supported by data and results, emphasizing the system's effectiveness in managing rainwater and its significance for wastewater systems.
This text has been added to the Discussion section:
„Efficient rainwater management is vital for urban wastewater systems facing flood and contamination risks. Our innovative Armlet with Wet Circuit Measurement System is tailored for enclosed conduits like sewage and stormwater channels. Precision engineering replicates closed channel conditions, aided by advanced materials and sensors for real-time rainwater detection. Our data-driven approach, rooted in rigorous lab experiments, explores various armlet shapes and sizes to minimize interference with wastewater flow. Accurate detection and quantification of rainwater enable proactive maintenance, averting floods and environmental harm. This innovative solution extends beyond monitoring, addressing rainwater-related challenges, enhancing urban wastewater management efficiency and sustainability”.
We thank the reviewer for their valuable input, which will help us improve the clarity and comprehensibility of our manuscript.
Round 2
Reviewer 1 Report
Thanks for addressing all my comments. I think the current version is good to be published.